# On Joint Learning for Solving Placement and Routing in Chip Design

**Ruoyu Cheng**  **Junchi Yan**[*]
Department of Computer Science and Engineering
MoE Key Lab of Artificial Intelligence, AI Institute
Shanghai Jiao Tong University, Shanghai, China, 200240
{roy_account,yanjunchi}@sjtu.edu.cn

## Abstract

For its advantage in GPU acceleration and less dependency on human experts, machine learning has been an emerging tool for solving the placement and routing problems, as two critical steps in modern chip design flow. Being still in its early stage, there are fundamental issues: scalability, reward design, and end-to-end learning paradigm etc. To achieve end-to-end placement learning, we first propose a joint learning method termed by DeepPlace for the placement of macros and standard cells, by the integration of reinforcement learning with a gradient based optimization scheme. To further bridge the placement with the subsequent routing task, we also develop a joint learning approach via reinforcement learning to fulfill both macro placement and routing, which is called DeepPR. One key design in our (reinforcement) learning paradigm involves a multi-view embedding model to encode both global graph level and local node level information of the input macros. Moreover, the random network distillation is devised to encourage exploration. Experiments on public chip design benchmarks show that our method can effectively learn from experience and also provides intermediate placement for the post standard cell placement, within few hours for training.

## 1 Introduction

With the breakthrough of semiconductor technology, the scale of integrated circuit (IC) has surged exponentially, which challenges the scalability of existing Electronic Design Automation (EDA) algorithms and technologies. Placement is one of the most crucial but time-consuming steps of the chip design process. It maps the components of a netlist including macros and standard cells to locations on the chip layout, where standard cells are basic logic cells e.g. logic gates and macros are functional blocks e.g. SRAMs. A good placement leads to better chip area utilization, timing performance and routability. Based on the placement assignment, routing assigns wires to connect the components, which is strongly coupled with placement task. In addition, the placement solution also serves as a rough estimation of wirelength and congestion, which is valuable in guiding the earlier stages of design flow. The objective of placement is to minimize metrics of power, performance, and area (PPA) without violating the constraints such as placement density and routing congestion.

We provide a pipeline to the placement problem: the input of placement is a netlist represented by hypergraph $H = (V, E)$, where $V$ denotes set of nodes (cells) and $E$ denotes set of hyperedges (nets) that indicates the connectivity between circuit components. Marco placement firstly determines the locations of macros on the chip canvas, followed by immense numbers of standard cells adjust their position based on adjacent macros and finally obtains the full placement solution, as shown in Fig. 1(a). The routing problem, however, takes placement solution as input and tries to connect

---

[*]Correspondence author is Junchi Yan.

those electronic components in a circuit coarsely. Without violating constraints on edges between neighboring tiles, the target of routing is to minimize the total wirelength, as shown in Fig. 1(b).

In this work, we first propose an end-to-end learning approach DeepPlace for placement problem with two stages. The deep reinforcement learning (DRL) agent places the macros sequentially, followed by a gradient-based optimization placer to arrange millions of standard cells. In addition, we develop a joint learning approach DeepPR (i.e. deep place and routing) for both placement and the subsequent routing task via reinforcement learning. The main contributions of this paper are:

1) For learning based placement, we propose an end-to-end approach DeepPlace for both macros and standard cells, whereby the two kinds of components are sequentially arranged by reinforcement learning and neural network formed gradient optimization, respectively. To our best knowledge, this is the first learning approach for the joint placement solving of macro and standard cells. In the previous works, these two tasks are independently solved either for macro [1] or standard cells [2].

2) We also propose DeepPR to jointly solve placement and routing via (reinforcement) learning, which again to our best knowledge, has not been attempted in literature before.

3) To adapt reinforcement learning more effectively into our pipeline, we design a novel policy network that introduces both CNN and GNN to provide two views to the placement input, in contrast to previous works that use CNN [3] or GNN [1] alone to obtain the embedding. The hope is that both global embedding and node level embedding information can be synthetically explored. We further adopt the random network distillation [4] to encourage exploration in reinforcement learning.

4) We provide experimental evaluation for both joint macro/standard cell placement and joint placement & routing. Results show that our method surpasses the separate placement and routing pipeline by a notable margin. Code partly public available at: https://github.com/Thinklab-SJTU/EDA-AI.

## 2   Related Work

**Classical methods for Placement.** The history of global placement can trace back to the 1960s [5, 6]. A wide variety of methods have been proposed since then, most of which fall into three categories: partitioning-based methods, stochastic/hill-climbing methods, and analytic solvers.

In early years, partition-based methods adopted the idea of divide-and-conquer: netlist and chip layout are partitioned recursively until the sublist can be solved by optimal solvers. This hierarchical structure makes them fast to execute and natural to extend for larger netlists, while scarifying the solution quality since each sub-problem is solved independently. Some multi-level partitioning methods [7] were developed afterwards. Stochastic and hill-climbing, as the second category, are mainly based on annealing methods [8] which is inspired from annealing in metallurgy that involves heating and controlled cooling for optimal crystalline surfaces. In practice, simulated annealing (SA) optimizes a given placement solution by random perturbations with actions such as shifting, swapping and rotation of macros [9, 10]. Although SA is flexible and able to find the global optimum, it is time-consuming and hard to deal with the ever-increasing scale of circuit.

For the analytic solvers, force-directed methods [11] and non-linear optimizers [12, 13] are commonly adopted. In comparison, the quadratic methods are computational efficient but showing relatively lower performance, while non-linear optimization approximates the cost functions more smoothly with the cost of higher complexity. Recently, however, modern analytical placers such as ePlace [14] and RePlAce [15] introduce electrostatics-based global-smooth density cost function and Nesterov's method nonlinear optimizer that achieve superior performance on public benchmarks. They formulate each node of the netlist as positively charged particle. Nodes are adjusted by their repulsive force and the density function corresponds to system's potential energy. These analytical methods update positions of cells in gradient based optimization scheme and generally can handle millions of standard cells by parallelization on multi-threaded CPUs using partitioning to reduce the running time.

Nevertheless, all methods mentioned above perform heavy numerical computation for large-scale optimization problem on the CPUs, which lacks exploration of GPU's opportunity. DREAMPlace [2] is inspired by the idea that the analytical placement problem is analogous to training a neural network. They both involve optimizing parameters and minimizing a cost function. Based on the state-of-the-art analytical placement algorithm RePlAce, DREAMPlace implements hand-optimized key operators by deep learning toolkit PyTorch and achieves over $30\times$ speedup against CPU-based tools.

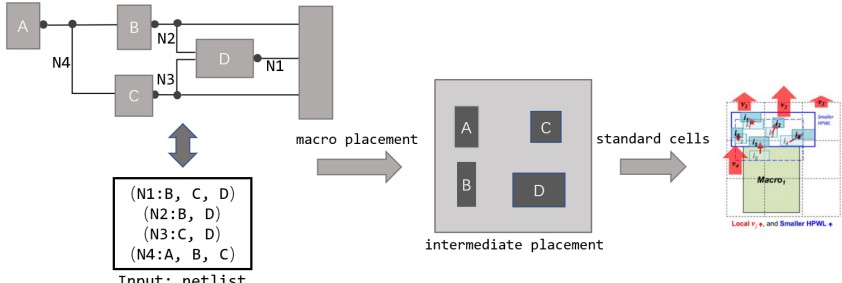

(a) **Introduction to placement pipeline.** Based on the input – a netlist with node features, we are supposed to determine positions of macros as well as standard cells on a chip canvas.

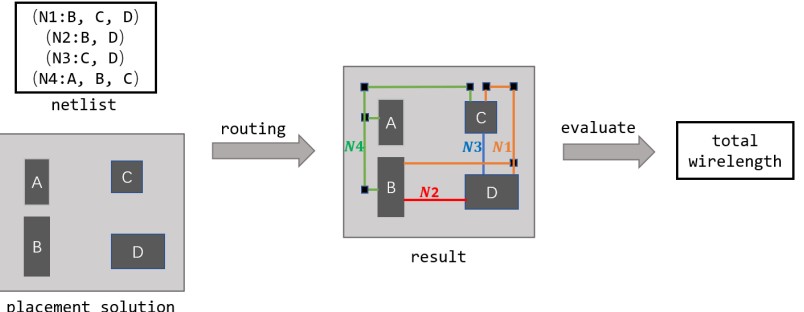

(b) **Introduction to routing pipeline.** Once the placement result has been obtained, routing assigns the wires to connect components in each net such as $N_1$ and $N_2$.

Figure 1: Illustration for concepts and the pipeline involved in placement and routing problems.

**Learning-based methods for Placement.** Recently learning-based methods for placement especially reinforcement learning (RL) have been proposed to obtain the generalization ability. Google [1] proposes the first end-to-end learning method for placement of macros that models chip placement as a sequential decision making problem. In each step, the RL agent places one macro and target metrics are used as reward until the last action. GNN is adopted in the value network to encode the netlist information and deconvolution layers in the policy network output the mask of current macro position. Another line is to solve the placement problem by combining RL and heuristics [16]. They propose a cyclic framework between the RL and SA, where RL module adjusts the relative spatial sequence between circuit components and SA further explores the solution space based on RL initialization. In comparison, the former one is a learning based approach with the same objective function as analytic solvers, while the later is essentially an annealing solver.

**Classical & Learning-based methods for Routing.** Global routing generally begins with decomposing a multiple-pin net problem into a set of two-pin connection problems [17, 18, 19]. After that, each pin-to-pin routing problem is solved by classical heuristic-based router such as rip-up and reroute [20], force-directed routing [21] and region-wise routing. Recently, machine learning techniques have been applied for routing information prediction, including routing congestion [22], the routability of a given placement [23] and circuit performance [24]. Meanwhile, RL method has also been proposed to handle routing problems. [25] uses a DQN model to decide actions of the routing direction, i.e. going north, south, etc at each step. [26] proposes an attention-based REINFORCE algorithm to select routing orders and use pattern router to generate actual route once routing order is determined. While [27] applies genetic algorithm to create initial routing candidates and uses RL to fix the design rule violations incrementally.

Different from current learning-based placer such as [1] that groups the standard cells into a few thousand clusters and uses force-directed methods for coarse placement solution, our work combines reinforcement learning with a gradient based optimization scheme without clustering to obtain the full placement solution through end-to-end learning scheme. While [15] considers a local density function as penalty factor to address the requirement for routability, the proposed method, to our best knowledge, is the first attempt to jointly solve placement and routing via reinforcement learning.

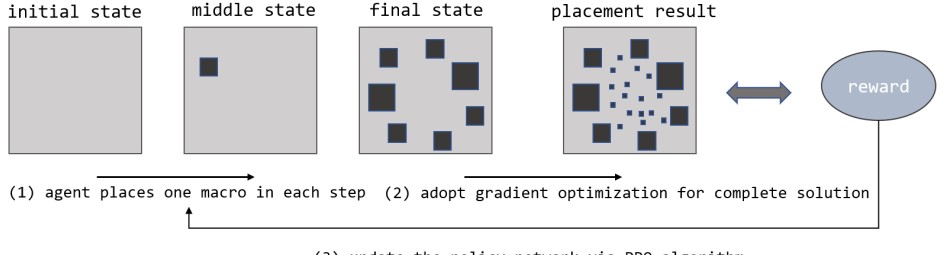

(a) Diagram of our learning-based placer for arranging both macros and standard cells (**DeepPlace**)

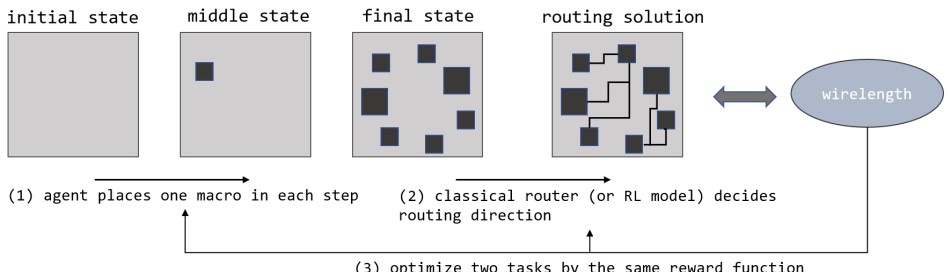

(b) The joint learning approach to solve placement and routing via reinforcement learning (**DeepPR**)

Figure 2: Overview of how our RL agent involves in two subsequent tasks. We term our two methods by DeepPlace (for placement only) and DeepPR (i.e. deep placement and routing), respectively.

## 3 Methodology

### 3.1 Problem Formulation

First of all, we target the problem of macro placement, whose objective is to determine locations of macros on the chip canvas with no overlap and wirelength minimized. Our RL agent sequentially maps the macros to valid positions on the layout. Once all macros have been placed, we either fix their positions and adopt gradient-based placement optimization to obtain a complete placement solution with corresponding evaluation metrics such as wirelength and congestion, as shown in Fig. 2(a). Or alternatively, we adopt classical router or another RL agent to route the placement solution and regard the exact total wirelength as rewards for both placement and routing task, which is shown in Fig. 2(b). The key elements of the Markov Decision Processes (MDPs) are defined as follows:

- **State** $s_t$**:** the state representation consists of two parts, global image $I$ portrayed the layout and netlist graph $H$ which contains detailed position of all macros that have been placed. Note that placement problem is analogous to a board game (i.e., chess or GO), both of which are required to determine the position of pieces (macros). Therefore, we model the chess board (chip canvas) as a binary image $I$ where 1 denotes position that has been occupied. In addition, the rule of board game is similar to netlist graph $H$, which is complementary to the global image.

- **Action** $a_t$**:** the action space contains available positions in the $n \times n$ canvas at time $t$, where $n$ denotes the size of grid. Once a spare position $(x, y)$ is selected by the current macro, we set $I_{xy} = 1$ and remove this position from the available list.

- **Reward** $r_t$**:** the reward at the end of episode is a negative weighted sum of wirelength and routing congestion from the final solution. The weight is a trade-off between main objective wirelength and routing congestion which indicates the routability for routing task. Different from other deep RL placers that set the reward to 0 for all previous actions, we adopt random network distillation (RND) inspired from [4] to calculate intrinsic rewards at each time step.

The policy network learns to maximize the expected reward from placing prior chips and improves placement quality over episodes, which is updated by Proximal Policy Optimization (PPO) [28].

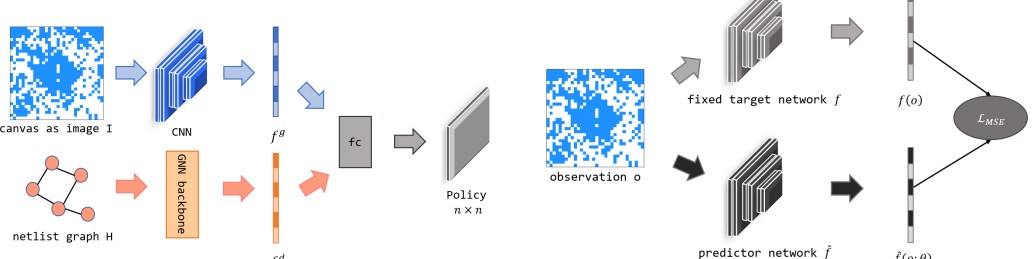

(a) Our policy network architecture        (b) The random network distillation module

Figure 3: **Policy network architecture and random network distillation module for end-to-end placement (and routing).** (a) The CNN network takes global image $I$ (as a representation for current placement) as input and generates the global embedding $f^g$, while detailed node embedding $f^d$ is obtained from a GNN backbone. The policy network outputs the distribution over available positions after concatenation and $fc$ layer. (b) Randomly initialized target network and predictor network take observation $o$ to generate an embedding $f(\text{o})$ and $\hat{f}(\text{o}; \theta)$ respectively. The expected MSE between two embeddings is used as intrinsic reward as well as loss function for predictor network in each step.

## 3.2 The Structure of Policy Network

Since both solving placement problem and playing board game such as GO are essentially to determine the position of macros (pieces) in a sequential manner, we model the current state as an image $I$ of size $n \times n$. We select $n$ from $\{32, 64\}$ in the experiments. $I_{xy} = 1$ when previous macro is placed on position $(x, y)$. This image representation gives an overview of the partial placement with loss of some detailed information. We further obtain the global embedding from convolutional neural network (CNN) which shows significant performance in GO game [29]. Moreover, the netlist graph as critical input information implies the rule of reward calculation, which is detailed guidance on the action prediction. In this case, we develop a graph neural network (GNN) architecture that produces detailed node embedding for current macro in consideration. The role of graph neural networks is to explore the physical meaning of netlist and distill information about connectivity of nodes into low-dimensional vector representations which is utilized in following calculations. After we obtain both global embedding from CNN and detailed node embedding from GNN, we fuse them by concatenation and pass the result to a fully-connected layer to generate a probability distribution over actions. We believe the multi-view embedding model is able to synthetically explore global and node level information. The whole structure of policy network is shown in Fig. 3(a).

## 3.3 Reward Design

### 3.3.1 Extrinsic Reward Design

Although the exact target of chip placement is to minimize power, performance and area, it takes industry-standard EDA tools several hours to evaluate, which is unaffordable for RL agent that needs thousands of examples to learn. It is natural to find approximate reward functions that positively correlated with true reward. We define cost functions with both wirelength and congestion, trying to optimize the performance and routability simultaneously:

$$R_E = -Wirelength(P, H) - \lambda \cdot Congestion(P, H) \tag{1}$$

where $P$ denotes the placement solution, $H$ denotes the netlist and $\lambda$ is a hyperparameter that weights the relative importance according to the settings of a given chip.

**Wirelength.** As the true wirelength is actually decided by routing, we employ half-perimeter wirelength (HPWL) as the approximation for wirelength. The definition of HPWL is half-perimeter of the bounding boxes for all nodes in the netlist, which can be roughly assumed as the length of its Steiner tree (the infimum bound of routing). For a given net $n_i$ with the set of end points $\{x_i\}$ and $\{y_i\}$, the HPWL can be calculated by:

$$HPWL(n_i) = (MAX\{x_i\} - MIN\{x_i\} + 1) + (MAX\{y_i\} - MIN\{y_i\} + 1) \tag{2}$$

**Congestion.** The idea of congestion comes from the fact that each routed net will occupy a certain amount of available routing resources. We adopt Rectangular Uniform wire Density (RUDY) [30] to

approximate routing congestion as HPWL is an intermediate result during the calculation process, and accuracy of RUDY is relatively high. We set the threshold for routing congestion as $0.1$ in our experiments, otherwise it causes congestion check failure.

### 3.3.2   Intrinsic Reward Design

Note that no useful signal can be provided until the end of a trajectory. It makes the placement a sparse reward task. Episodic reward is often prone to make algorithms get stuck during the training process and suffer from inferior performance and inefficient sample complexity. Inspired by the idea of random network distillation (RND) [4], we give an intrinsic bonus in each time step to encourage exploration as shown in Fig. 3(b). There are two networks involved in RND: a fixed and randomly initialized target network and a predictor network trained on global images $I$ collected by the agent. Given the current observation $o$ (which also refers to the global image), the target network and predictor network generate an embedding $f(o)$ and $\hat{f}(o; \theta)$ respectively. The intrinsic reward is:

$$R_T = \|\hat{f}(o; \theta) - f(o)\|^2 \tag{3}$$

After that, the predictor network is trained by SGD to minimize this expected MSE which distills randomly initialized network into a trained one. This distillation error could be seen as a quantification of prediction uncertainty. As a novel state is expected to be dissimilar to what the predictor has been trained on, the intrinsic reward also becomes higher to encourage visiting novel states.

### 3.4   Combination with Gradient-based Placement Optimization

The placement for millions of standard cells is as important as placing the macros, considering its influence on the evaluation metrics for macro placement. To ensure the runtime of each iteration affordable for training, we apply state-of-the-art gradient based optimization placer DREAMPlace [2] to arrange standard cells in the reward calculation step. On the one hand, the position of large macros as fixed instances will influence the solution quality of gradient based optimization placer, which can improve over time through training. On the other hand, better approximation to the metrics such as wirelength leads to a better guidance for training the agent. As a result, the combination of RL agent with gradient based optimization placer will mutually enhance each other. Furthermore, the state-of-the-art tool DREAMPlace implements key kernels in analytical placement, e.g., wirelength and density computation with deep learning toolkit, which fully explores the potential of GPU acceleration and reduces the runtime in less than a minute.

### 3.5   Joint Learning of Placement and Routing

Routability is one of the most critical factors to consider during placement, hence routing congestion is a necessary component in the objective (reward) function in most previous methods. However, congestion as an implicit routability model is rough and not always accurate. Meanwhile, HPWL as proxy of wirelength also introduces bias towards the true target. This motivates us to jointly learn placement and routing task, both of which try to minimize the wirelength in practice. We can adopt any routing method in DeepPR including RL model as well as classical router to determine the routing direction after decomposing the netlist obtained from the placement task to pin-to-pin routing problems. The overall wirelength is then used as episodic reward for both placement and routing agents to optimize two tasks respectively. The advantages of this joint learning paradigm are twofold. On one hand, placement solution provides abundant training data for the routing agent, instead of randomly generated data used in previous work which lacks of modeling the distribution of real domain data. On the other hand, routing provides a direct objective for the placement agent to optimize, hence relieving the need of intermediate cost models and reducing bias in the reward signal.

## 4   Experiments

### 4.1   Benchmarks and Settings

We perform experiments based on the well-studied academic benchmark suites ISPD-2005 [31], which has not been investigated by learning-based approaches before, to our best knowledge. To demonstrate the benefit from our end-to-end placement learning of both macros and standard cells, we embody the same designs as the ISPD-2005, except that most fixed macros are exchanged to movable ones for the agent to organize. Parameters of all the eight circuits in ISPD-2005 benchmark

Table 1: Statistics for our modified ISPD-2005 benchmark suites. RL agent only places those movable macros, and design density is the ratio of the area sum of total objects over area of placement region.

| Circuits | # Total Cells | # Nets | # Movable Macros | # Fixed Macros | Design Density |
|---|---|---|---|---|---|
| adaptec1 | 211K | 221K | 514 | 29 | 75.7% |
| adaptec2 | 255K | 266K | 542 | 24 | 78.6% |
| adaptec3 | 451K | 466K | 710 | 13 | 74.5% |
| adaptec4 | 496K | 516K | 1309 | 20 | 62.7% |
| bigblue1 | 278K | 284K | 551 | 9 | 54.2% |
| bigblue2 | 558K | 577K | 948 | 22136 | 37.9% |
| bigblue3 | 1097K | 1123K | 1227 | 66 | 85.6% |
| bigblue4 | 2177K | 2230K | 659 | 7511 | 44.3% |

after modification are summarized in Table 1. This new benchmark is able to serve as an appropriate environment for evaluating algorithms for the macro placement. We also study two large-scale dataset bigblue3 and bigblue4 which include millions of cells to verify the scalability of our proposed method. It is worth noting that our method can handle flexible number of movable/fixed macros as well.

We use PPO in [28] for all the experiments implemented with Pytorch [32], and the GPU version of DREAMPlace [2] is adopted as gradient based optimization placer for arranging standard cells. We adopt GCN [33] as the GNN backbone which consists of two layers with 32 and 16 feature channels, respectively. The Adam optimizer [34] is used with $2.5 \times 10^{-4}$ learning rate. All experiments are run on 1 NVIDIA GeForce RTX 2080Ti GPU, and parallelism is not considered in the experiment. Since our reinforcement learning environment is deterministic, the error bars can be neglected.

## 4.2   Pretraining

Our goal is to train a reinforcement learning agent that can generate high-quality intermediate placement for the post standard cell placement task as well as subsequent routing task. However, training the end-to-end learning paradigms from scratch may suffer from longer time for convergence or even worse final performance, as the majority of the runtime is taken by gradient based optimization placer in each episode. In addition, the initial policy network starts off with random weights and generates low-quality intermediate placement, which needs more steps for exploration in the immense state space. Our intuition is that an agent capable of minimizing the wirelength only containing those macros should be good enough to provide an intermediate placement. Therefore, we propose to train the policy network by a partial netlist which only consists of connectivity between macros, and then finetune this pre-trained policy for subsequent tasks. The idea is similar to curriculum learning that gradually introduces more difficult task to speed up training.

## 4.3   Results on Joint Macro/Standard cell Placement

As mentioned previously, our RL agent generates intermediate macro placement and then adopts gradient-based optimization placer to obtain complete placement solution, while utilizing the pre-trained policies training with partial netlist consisting of macros only. It is worth noting that minimizing the wirelength between macros (i.e., the pre-training task) will not directly lead to global minimum of full placement, thus to what extent we pretrain the agent needs to consider attentively. Fig. 4 shows the convergence plots for agents training from scratch versus from pre-trained policy for different timesteps on circuit $adaptec1$. The agent pre-trained for 300 iterations achieves the best result, while training from scratch tends to be less stable and performs the worst. This may due to the increasing complexity of reward function, so that it is hard for agent to predict performance of standard cell placement afterwards. Although the network pre-trained for 500 iterations starts with a higher reward, it fails to obtain a better placement, likely because excessive pre-training hurts exploration and is prone to overfitting in terms of true reward function. Hence we finetune the policy based on moderate pre-training in further experiments.

In Table 2 we compare the total wirelength of our end-to-end learning approach DeepPlace with sequential learning placer that applies gradient based optimization after pre-training separately, as well as DREAMPlace which arranges movable macros by heuristic in the beginning of optimization. Both DeepPlace and sequential learning placer consistently generate viable placements, while DREAMPlace fails to meet congestion criteria in most cases for overlap between macros, which shows the advantage of our RL agent for macro placement. Our end-to-end approach DeepPlace

Table 2: **Comparison on macro/standard cell placement task on circuits from ISPD-2005.** Performance w.r.t. wirelength for three methods: 1) our end-to-end DeepPlace, 2) sequential placer (improving [1] for macro and using DREAMPlace for standard cell placement, separately) and 3) apply DREAMPlace directly on both macro and standard cell by treating them as general cells.

| Circuits | DeepPlace (ours) | Sequential Learning placer | Analytic Placer DREAMPlace* |
|---|---|---|---|
| adaptec1 | **80117232** | 86675688 | 83749616 |
| adaptec2 | **123265964** | 124125088 | congestion check failure** |
| adaptec3 | **241072304** | 257956784 | 264789408 |
| adaptec4 | **236391936** | 255770672 | congestion check failure |
| bigblue1 | **140435296** | 168456768 | congestion check failure |
| bigblue2 | **140465488** | 142037456 | congestion check failure |
| bigblue3 | **450633360** | 460581568 | congestion check failure |
| bigblue4 | **951984128** | 1039976064 | congestion check failure |

* Note RePlace [15] is supposed to always achieve the same wirelength as DREAMPlace (see also in [2]) because they only differ in the optimization schedule and DREAMPlace speeds up RePlace.
** The final macro/standard cell placement result fails to meet congestion criteria as discussed in Sec. 3.3.1, which is probably due to overlap between macros. In our analysis, the reason may be that DREAMPlace is not suited for macro placement as it directly searches for x-y position in continuous space while the other two solvers applies RL for macro placement whose action is discrete by nature with some protection against overlap.

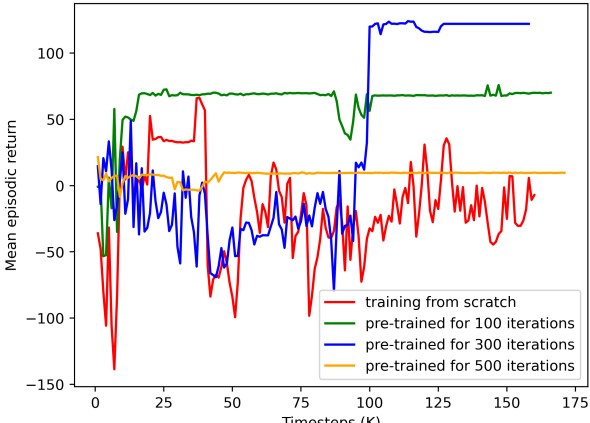

Figure 4: Comparison of agents pre-trained on circuit $adaptec1$. Mean episodic return (i.e., the extrinsic reward with both wirelength and congestion) increases as the iterations of pre-training increase in a certain range, while excessive pre-training hurts exploration.

also outperforms the sequential one in all test cases, demonstrating the advantage of joint learning paradigm. In Fig. 5 we further compare the runtime of DeepPlace with classical placer RePlace on eight circuits from ISPD-2005 benchmark. We can see that the runtime of RL agent for macro placement grows much slower than gradient based optimization placer for standard cells, and our total runtime achieves approximately $4\times$ speedup over RePlace on various circuits. Finally, the macro/standard cell placement results are visualized in Fig. 6.

## 4.4 Results on Joint Placement and Routing

We compare joint learning approach DeepPR to the separate pipeline of placement and routing. We evaluate our method on the same circuits as mentioned above, except that all standard cells are omitted for convenience. The strategy of pre-training is adopted in the experiments as well to speed up the training procedure and reach a higher reward, since training from scratch requires to collect more experience. The evaluation of final wirelength after routing are shown in Table 3. It shows that the reinforcement learning agent dramatically improves the placement quality of macros with routing wirelength as the feedback in all test cases, compared with the separate pipeline. The significant difference between DeepPR and the separate pipeline indicates that HPWL as a commonly used objective in placement is not accurate enough to approximate the exact total wirelength after routing. Meanwhile, our joint learning scheme is able to provide unbiased reward signal by optimizing directly towards final metrics, which is potential to further increase the performance of current placers.

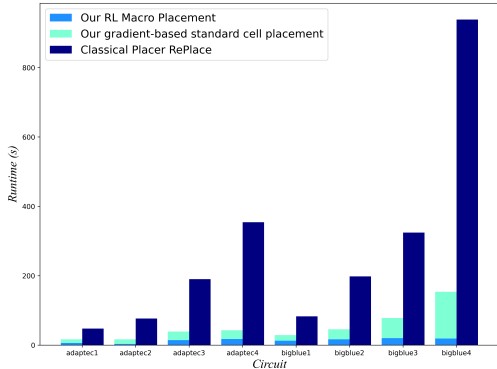

Figure 5: Runtime breakdown and comparison with classical placer on eight circuits from ISPD-2005. In our method DeepPlace, the RL module only occupies a small percentage of runtime for larger circuit, and externally achieves approximately $4\times$ speedup over the classic placer RePlace [15]. Time cost of sequential learning placer is the same as DeepPlace, as their only difference lies in training protocol while model structure is the same. As shown in Table 2, some results produced by RePlace are faced with high congestion problem due to overlap between macros.

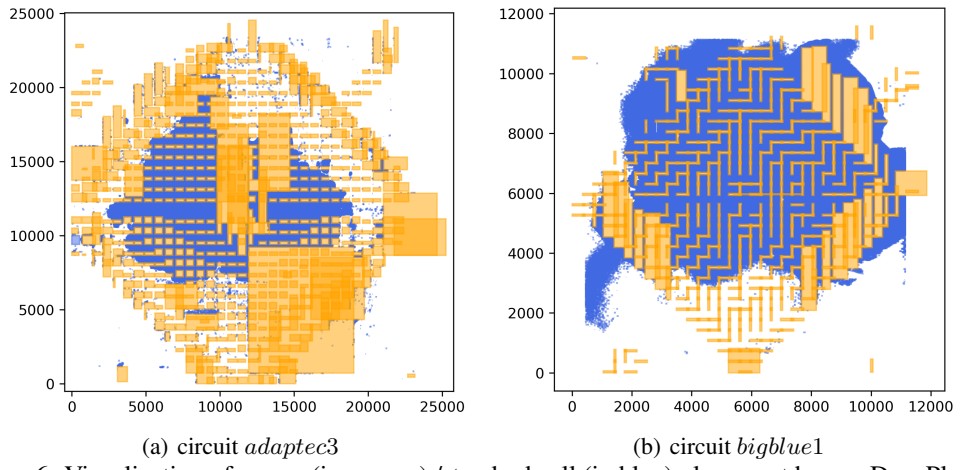

(a) circuit $adaptec3$          (b) circuit $bigblue1$

Figure 6: Visualization of macro (in orange) /standard cell (in blue) placement by our DeepPlace on circuits $adaptec3$ and $bigblue1$ from ISPD-2005. Our RL agent tends to place macros in each net close to each other, and searches the nearest valid position to replace if two macros overlap, which perhaps contributes to the final formation of a diamond-shaped placement.

Table 3: Final wirelength comparison for joint placement and routing problem on ISPD-2005 benchmark. Note time cost can be regarded as the same for inference stage as the difference of the two compared methods are only for their training protocol while the model structure is the same.

| Method | adaptec1 | adaptec2 | adaptec3 | adaptec4 | bigblue1 | bigblue2 | bigblue3 | bigblue4 |
|---|---|---|---|---|---|---|---|---|
| DeepPR (ours) | **5298** | **22256** | **32839** | **63560** | **8602** | **16819** | **94083** | **17585** |
| Separate Pipeline | 6077 | 23725 | 37029 | 79573 | 9078 | 17849 | 98480 | 17829 |
| time (s) | 10.63 | 10.66 | 40.96 | 98.23 | 16.12 | 32.71 | 77.60 | 18.73 |

## 4.5 Ablation Study

To better verify the effectiveness of our multi-view policy network that introduces both CNN and GNN and the random network distillation module in reward function, we conduct ablation studies on the pretraining task using modified environment (circuit) $adaptec1$. In Fig. 7 we compare our

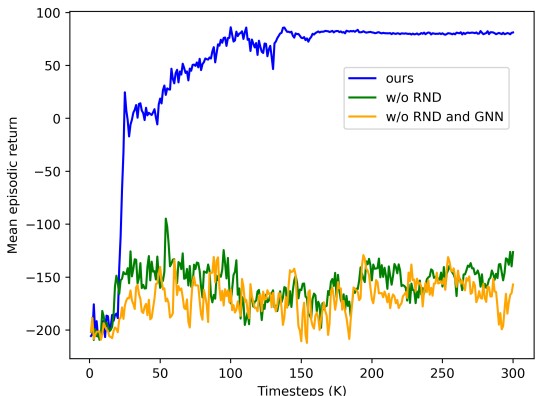

Figure 7: Mean episodic return comparison: the baseline global embedding from CNN, combined embedding from both CNN and GNN, and our agent with additional intrinsic reward from RND. Our agent significantly outperforms baselines on circuit $adaptec1$ (so for other circuits).

proposed agent with the agents without RND module as well as detailed node embedding from GNN. It is not surprising that the baseline method only shows modest improvement over the initial random policy, due to the sparse reward signal and immense possible placements. Note that the performance of using CNN or GNN alone to obtain the embedding is similar, hence we present one of them in the figure. Combination of global embedding and detailed node embedding makes it possible to learn rich representations of the state space, which achieves better performance than baseline structure. Nevertheless, our placer with RND to provide intrinsic reward for exploration in each step surpassed the baseline by a large margin, which is consistent with our discussion in Sec. 3.3.2.

## 5   Conclusion and Outlook

We have presented an end-to-end learning paradigm for placement and routing, which are two of the most challenging and time-consuming tasks in chip design flow. For macro placement, we adopt the reinforcement learning while for the more tedious standard cell placement, we resort to the gradient based optimization technique which can be more cost-effective, and successfully incorporate it into the end-to-end pipeline. One step further, we develop reinforcement learning for joint solving placement and routing, which has not been studied before. In particular, a two-view based embedding model is developed to fuse both global and local information and distillation is devised to improve exploration. Experimental results on public benchmark show the effectiveness of our solver.

For future work, we plan to explore learning the placement of macros and standard cells together with routing for macros and standard cells in an end-to-end paradigm, since these two tasks are strongly coupled in the chip design process. However, no existing work has fulfilled this goal to our knowledge, due to its large scale and complexity.

**Limitation.** Our model currently is limited to a moderate number of macros for placement, due to the scalability issue of GNN and the large action space for reinforcement learning. Another improvement for future work is to take the size and shape of macros into consideration to further reduce overlap between huge macros, as well as combine human heuristics and rules into the learning paradigm.

**Potential negative societal impacts.** Our method facilitates the automated placement and routing, while in the EDA industry there are many staff working on chip design and our software may put some of them in a situation for being unemployed.

## Acknowledgments and Disclosure of Funding

This work was partly supported by National Key Research and Development Program of China (2020AAA0107600), Shanghai Municipal Science and Technology Major Project (2021SHZDZX0102), and Huawei Inc.

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
