# OpenReview forum: "On Joint Learning for Solving Placement and Routing in Chip Design"
_NeurIPS.cc/2021/Conference — NeurIPS 2021 Poster_

### Official Review · Reviewer_NVyt · 2021-07-11

**Rating:** 7
**Confidence:** 4

**Summary:**

This paper proposed a joint learning method for the placement of macros and standard cells, by the integration of reinforcement learning with a gradient based optimization. The paper also proposed to jointly solve placement and routing via reinforcement learning, which has been the first time attempted in literature. The designed policy network incorporates both CNN and GNN to provide a multi-view embedding with a global embedding and a node level embedding. The paper also proposed a random network distillation to encourage exploration. Empirically, the method surpasses the separate placement and routing pipeline by a notable margin.

**Limitations And Societal Impact:**

Both limitations and societal impact are sufficiently addressed.

**Main Review:**

Strengths:
1. The paper is clearly, well written. Explanations on placement and routing problems, graphical illustration are good. The paper provides a thoroughly written related work section.

2. The proposed method obtains the full placement solution through end-to-end learning without clustering. Jointly solving placement and routing via reinforcement learning is new. The benefits are pretty clear: first, the placement solutions provides abundant training data for the routing agent, instead of relying on randomly generated data used in previous work. Routing, on the other hand, provides a direct objective for the placement agent to optimize, hence mitigating the reliance on an intermediate cost models and reducing bias in the reward.

3. The placement involves the placement of macros (there are a few of them) and standard cells (there are many of them), which makes the solution more significant than solution [1]. The reviewer applauds the efforts making this possible.

Weaknesses:
1. The advantage of combing CNN and GNN is small, as indicated in Figure 4.

2. RND improves the performance significantly, but can be replaced with related methods such as entropy based methods or regularization based methods. The paper does not provide ablation study on that.

3. There are not sufficient benchmarks. Only two circuits are provided. There is a lack of details on those circuits.

4. There are not sufficient baselines to compare with. The reviewer understands that it could be a comparatively new domain, and not many related works are available. But the reviewer would be glad to see some comparisons with work [1] by reimplementing [1] in your setting. Also, additional results compared to human placement baseline is highly recommended.


Detailed comments:
1. [1] approach is not a analytic solver or via gradient updates. It is still a reinforcement learning approach, using a transferable value network.

2. There isn't a ablation study demonstrating the necessity of multi-view embedding. Comparisons of methods with and without global or node level information should be provided.

3. How does the intrinsic reward based method via random network distillation compare to a entropy based method such as SAC [2]?

4. Why only two layers of GNN is used? It seems to small for a problem like placement and routing.

Overall, the reviewer likes the paper about its novelty in combing RL and gradient-based methods, jointly optimizing placement of macros and standard cells end to end, and jointly optimizing placement and routing, the proposal of pretraining on a simpler proxy task. However, the downside is its insufficient comparisons with related work. The paper needs some stronger baselines.


[1] Chip placement with deep reinforcement learning, https://arxiv.org/abs/2004.10746
[2] Soft Actor-Critic: Off-Policy Maximum Entropy Deep Reinforcement Learning with a Stochastic Actor
, https://arxiv.org/abs/1801.01290

**Time Spent Reviewing:**

2

---

> ### Author Response · Authors · 2021-08-10
> **Rebuttal to Reviewer NVyt**
>
> Thank you for the detailed feedback and your recognition to our work's novelty, clarity, potential influence and empirical strong performance. Below we respond to your comments.
>
> Responses to the main concerns:
>
> **Q1. *"The advantage of combing CNN and GNN is small, as indicated in Figure 4.***
>
> **Q2. *"RND improves the performance significantly, but can be replaced with related methods such as entropy based methods or regularization based methods. The paper does not provide ablation study on that.***
>
> **Q3.*"How does the intrinsic reward based method via random network distillation compare to a entropy based method such as SAC?"***
>
> **Q4.*"There isn't a ablation study demonstrating the necessity of multi-view embedding. Comparisons of methods with and without global or node level information should be provided."***
>
> **Re to Q1,2,3,4**: We verify the improvement of multi-view policy network and RND module in the final performance on circuit **adaptec3** from the benchmark ISPD-2005. Results are shown in the table below. Although the advantage of combing CNN and GNN is smaller than RND, the cost of adding a GNN by treating the netlist as graph is negeligible, while RND takes longer to train and compute. The entropy based method SAC shows compatible result when replacing the RND module, and we add the following comparison:
>
> |                  | Our Joint Learning Approach | w/o GNN | w/o RND | SAC   | Separate Pipeline |
> |------------------|-----------------------------|---------|---------|-------|-------------------|
> | Total wirelength | 32839                       | 34742   | 35556   | 33445 | 37029             |
>
>
> **Q5. *"There are not sufficient benchmarks. Only two circuits are provided. There is a lack of details on those circuits."***
>
> **Re**: Detailed statistics of circuits in our experiments are summarized in Table 1. We add new results of 3 additional circuits for ISPD-2005, including one large-scale dataset (bigblue3) that can make it more convincing. So far we have tested 5 out of 8 chips in ISPD-2005, and the remaining 3 chips will also be provided in the next few days if permited. The additional experimental results are shown in the tables below.
>
> First we demonstrate the statistics of three additional circuits from ISPD-2005 benchmark.
>
> | Circuits | # Total Cells | # Nets | # Movable Macros | # Fixed Macros | Design Density |
> |----------|---------------|--------|------------------|----------------|----------------|
> | adaptec3 | 451K          | 466K   | 710              | 13             | 74.5%          |
> | bigblue1 | 278K          | 284K   | 551              | 9              | 54.2%          |
> | bigblue3 | 1097K         | 1123K  | 1227             | 66             | 85.6%          |
>
> In table below we show results on Joint Macro/Standard cell placement, which is extension to Table 2. Note that DREAMPlace fails to meet congestion criteria in circuits bigblue1 and bigblue3 for overlap between macros, our approach and sequential learning placer consistently generate viable placements, which shows the advantage of our RL agent for macro placement.
>
> | Circuits | Our End-to-End Approach | Sequential Learning Placer (our immediate improvement to [1]) | Analytic Placer DREAMPlace |
> |----------|-------------------------|----------------------------|----------------------------|
> | adaptec3 | **241072304.0**             | 257956784.0                | 264789408.0                |
> | bigblue1 | **140435296.0**             | 168456768.0                | - (failed in legality)     |
> | bigblue3 | **450633360.0**             | 460581568.0                | - (failed in legality)     |
>
> Finally we present the results on Joint Placement and Routing, which is extension to Table 3.
>
> | Circuits  | Our Joint Learning Approach | Separate Pipeline |
> |----------|-----------------------------|-------------------|
> | adaptec3 | **32839**                       | 37029             |
> | bigblue1 | **8602**                        | 9078              |
> | bigblue3 | **94083**                       | 98480             |
>
> **Q6. *"There are not sufficient baselines to compare with. The reviewer understands that it could be a comparatively new domain, and not many related works are available. But the reviewer would be glad to see some comparisons with work [1] by reimplementing [1] in your setting. Also, additional results compared to human placement baseline is highly recommended."***
>
> **Re**: Actually the sequential learning placer in Table 2 refers to our immediate improvement of method [1]. A structured overview for the differences is shown as follows.
>
> |                                                                                  | targets for placement |                  | learning protocol               |                |
> |----------------------------------------------------------------------------------|-----------------------|------------------|---------------------------------|----------------|
> |                                                                                  | **macro cell**            | **standard cell**    | **reward**                          | **joint learning** |
> | our method                                                                       | RL                    | back-prop by DNN | based on cell placement result  | yes            |
> | Google's Nature 2021 [1]                                                         | RL                    | NA               | based on macro placement result | no             |
> | sequential learning placer for ablation study (our immediate improvement to [1]) | RL                    | back-prop by DNN | based on macro placement result | no             |
> | DREAMPlace                                                                       | heuristic             | back-prop by DNN | NA                              | no             |
>
> [1] Mirhoseini, Azalia, Anna Goldie, Mustafa Yazgan, Joe Wenjie Jiang, Ebrahim Songhori, Shen Wang, Young-Joon Lee et al. "A graph placement methodology for fast chip design." Nature 594, no. 7862 (2021): 207-212. (Note we cited its arxiv version in our paper as at that time the nature version has not been published yet)
>
>
> **Q7. *"[1] approach is not a analytic solver or via gradient updates. It is still a reinforcement learning approach, using a transferable value network."***
>
> **Re**: Thanks for your comments and we totally agree with you. Actually we intended to make comparsion between two existing RL placement methods at lines 95-96 in the paper. We will revise our writing to make it more precise and informative. [1] is a reinforcement learning approach indeed.
>
> **Q8. *"Why only two layers of GNN is used? It seems to small for a problem like placement and routing."***
>
> **Re**: For the placement and routing task, it is generally more concerned with the connectivity of nodes and there is often little node attribute for learning. We also tend to assume the GNN is used to extract the local information while the global information can be more effectively and efficiently extracted by our CNN. In fact, we did not find this important hyperparameter's setting in Google's Nature paper, and we provide an ablation study on circuit **adaptec3** to show two layers is cost-effective.
>
> | GNN layers   #  | 2     | 3     | 4     |
> |------------|-------|-------|-------|
> | wirelength | 32839 | 33486 | 32663 |

---

> > ### Comment · Reviewer_NVyt · 2021-08-24
> > **Good rebuttal**
> >
> > I would like to raise my score to 7 after reading the rebuttal. I think the authors well addressed my concerns and provided sufficient additional evidences.
> >
> > -My concern over too few evaluations and not much details about the circuits are addressed by three additional circuits.
> >
> > -"No comparison with state of the art" is addressed by pointing out the sequential RL is a re-implementation of Google's nature paper. It differentiate well from the Google's paper that this paper handles standard cells (while the Google's paper not) as well as macro blocks.
> > The reviewer is happy to find this paper is addressing something that the existing work does not address.
> >
> > -The combination of gradient based method with RL is novel and the evaluation is thorough. The story around tackling placement and routing in an end-to-end fashion but with different treatments is novel.

---

> > ### Public Comment · ~Changyong_Oh2 · 2023-10-11
> > **It seems that Google's work is also an end-to-end macro/std.cell placer**
> >
> > In the answer to Q6, the author provides a clear comparison between methods.
> >
> > The author argues that Google's paper is not an end-to-end placer because its reward does not take into account std cell placement according to the table given in the answer to Q6 above.
> >
> > In the arXiv version of Google's Nature paper [Chip Placement with Deep Reinforcement Learning
> > ](https://arxiv.org/pdf/2004.10746.pdf), it says that
> > > 3.3.7. POSTPROCESSING
> >
> > > To prepare the placements for evaluation by commercial EDA tools, we perform a greedy legalization step to snap
> > macros onto the nearest legal position while honoring the minimum spacing constraints. We then fix the macro placements and use an EDA tool to place the standard cells and evaluate the placement.
> >
> > Thus, they evaluate the placement after placing std cells using the force-directed method.
> > In other words, their reward also takes into account std.cell placement, too.
> >
> > Even though DeepPR does possess a novel end-to-end formalism in its consideration of routing, it appears that the end-to-end macro/std.cell placement was already proposed by Google.
> > The differences existing in DeepPlace in comparison with Google's work seems
> > - Using CNN in combination with GNN for transferable feature encoder
> > - Using extrinsic reward
> > - Using DreamPlace instead of force-directed method when placing std.cells
> > - etc...
> > - Supposedly, but not the first end-to-end macro/std.cell placer in the sense that end-to-end is defined by which information is reflected in the reward.
> >
> > Or am I misunderstanding the "end-to-end" macro/std.cell placement?
> >
> > Even though this is a kind of closed thread, I hope that the author or others interested in this paper can answer the question.

---

### Official Review · Reviewer_DTgM · 2021-07-16

**Rating:** 6
**Confidence:** 5

**Summary:**

The paper proposes a deep reinforcement learning method to solve the chip placement and routing problem. They first introduce an RL based method combined with a gradient-based method to approximate the wirelength. They then expand their RL agent to also support routing task as well.

**Limitations And Societal Impact:**

The authors explained the limitation of the work and it is possible societal impact.
- It would have been better if they could explain more in depth about the scalability limitation and the runtime of their method.

**Main Review:**

(I couldn't find any supplementary material nor appendix sections.)

Originality:
- The method is an extension on recent work on deep RL for placement. It also combines a quick placement method (DREAMPlace) for calculating RL reward. They were also the first to propose an RL agent for combined task of placement and routing.
- The paper provides a great literature review on the previous methods.

Quality:
- The authors provide some experimental results to support the claims, however, more experiments are required.
- They provided a limited ablation study for their first method (on RND and GNN).
- More experimental results are required, for example, the results in Figure 4 and 5, have to be an average of multiple runs with different seeds.
- The authors didn't report any information about the runtime of the method and the impact of transfer learning.
- The number of blocks in the evaluation are very limited. They could've used bigblue* blocks in the ISPD2005 as well.
- The existing results do not show the scalability of the proposed methods for real production blocks which include more cells.
- In Figure 4, Is the RND reward included in the y axis? If yes, then the comparison is meaningless, and if no, please explicitly mention what is y-axis.

Clarity:
- The paper is well written and organized.
- In Figure 1a, the top left image, what is the unnamed block on the right?
- Define acronyms before using them: SA, DRL.
- In Section 3.1, Action, It is not clear how macros that are larger than 1x1 are handled? How the grid size n is determined and why is it always square?
- How RUDY is calculated, since it is not part of DREAMPlace?
- A newer version of [1] can be find in: Mirhoseini, Azalia, Anna Goldie, Mustafa Yazgan, Joe Wenjie Jiang, Ebrahim Songhori, Shen Wang, Young-Joon Lee et al. "A graph placement methodology for fast chip design." Nature 594, no. 7862 (2021): 207-212.


Significance:
- The proposed methods are interesting and important for the community. It provides new methods to address the complex problem of placement and routing and will be likely interesting for the researchers and practitioners.

**Time Spent Reviewing:**

2

---

> ### Author Response · Authors · 2021-08-10
> **Rebuttal to Reviewer DTgM**
>
> Thank you for the detailed feedback and your recognition to our work for its potential influence, technical novelty and soundness, as well as overall writing and appropriate literature. Below we respond to your comments, and we will clarify the points in our new version.
>
> Responses to the questions:
>
> **Q1. *"In Figure 4, Is the RND reward included in the y axis? If yes, then the comparison is meaningless, and if no, please explicitly mention what is y-axis."***
>
> **Re**: No. The episodic return in y-axis refers to the target reward calculated in the end of an episode, i.e., cost functions with both wirelength and congestion. Intrinsic reward given by RND only serves as the intermediate reward.
>
> **Q2. *"In Figure 1a, the top left image, what is the unnamed block on the right?"***
>
> **Re**: The image is designed to help readers understand physical meaning of a netlist, so the circuit is actually virtual and incomplete. We can regard the unnamed block on the right as an output module.
>
> **Q3. *".. how macros that are larger than 1x1 are handled? How the grid size n is determined and why is it always square?"***
>
> **Re**: In our experiment, we set the size of all macros as $1\times1$. The grid size $n$ is a hyperparameter that trades off between runtime and the performance. Larger $n$ will consider macro placement in a more detailed manner, at the cost of longer runtime due to the growing action space. As most of the chip board in ISPD-2005 benchmark is square, we use a $n \times n$ chip canvas in formulation. The generalization to a $m \times n$ chip canvas is straightforward and easy to implement.
>
> **Q4. *"How RUDY is calculated, since it is not part of DREAMPlace?"***
>
> **Re**: As minimizing wirelength and congestion is also the objective of modern analytic placer, DREAMPlace implements these kernels for computing their gradients. Therefore, we make use of them directly from DREAMPlace. We will make it clear.
>
> **Responses to the main concerns:**
>
> **Q1. *".. more experiments are required."***
>
> **Q2. *"The number of blocks in the evaluation are very limited."***
>
> **Q3. *"The existing results do not show the scalability of the proposed methods for real production blocks which include more cells."***
>
> **Re**: We add new results of 3 additional circuits for ISPD-2005, including one large-scale dataset (bigblue3). So far we have tested 5 out of 8 chips in ISPD-2005, and the remaining 3 chips will also be provided in the next few days if permited. Additional experiment results are shown in the tables below.
>
> First we demonstrate the statistics of three additional circuits from ISPD-2005 benchmark.
>
> | Circuits | # Total Cells | # Nets | # Movable Macros | # Fixed Macros | Design Density |
> |----------|---------------|--------|------------------|----------------|----------------|
> | adaptec3 | 451K          | 466K   | 710              | 13             | 74.5%          |
> | bigblue1 | 278K          | 284K   | 551              | 9              | 54.2%          |
> | bigblue3 | 1097K         | 1123K  | 1227             | 66             | 85.6%          |
>
> Below we show results on Joint Macro/Standard cell placement, which is extension to Table 2. Note that DREAMPlace fails to meet congestion criteria in circuits bigblue1 and bigblue3 for overlap between macros, our approach and sequential learning placer consistently generate viable placements, which shows the advantage of our RL agent for macro placement.
>
> | Circuits | Our End-to-End Approach | Sequential Learning Placer (our immediate improvement to [1]) | Analytic Placer DREAMPlace |
> |----------|-------------------------|----------------------------|----------------------------|
> | adaptec3 | **241072304.0**             | 257956784.0                | 264789408.0                |
> | bigblue1 | **140435296.0**             | 168456768.0                | - (failed in legality)     |
> | bigblue3 | **450633360.0**             | 460581568.0                | - (failed in legality)     |
>
> [1] Mirhoseini, Azalia, Anna Goldie, Mustafa Yazgan, Joe Wenjie Jiang, Ebrahim Songhori, Shen Wang, Young-Joon Lee et al. "A graph placement methodology for fast chip design." Nature 594, no. 7862 (2021): 207-212. (Note we cited its arxiv version in our paper as at that time the nature version has not been published yet)
>
> Finally we present the results on Joint Placement and Routing, which is extension to Table 3.
>
> | Circuits  | Our Joint Learning Approach | Separate Pipeline |
> |----------|-----------------------------|-------------------|
> | adaptec3 | **32839**                       | 37029             |
> | bigblue1 | **8602**                        | 9078              |
> | bigblue3 | **94083**                       | 98480             |
>
> **Q4. *"Define acronyms before using them: SA, DRL."***
>
> **Re**: SA stands for simulated annealing and DRL for deep reinforcement learning. We will make it clear.
>
> **Q5. *"The authors didn't report any information about the runtime of the method and the impact of transfer learning."***
>
> **Re**: The training time in the all experiments is under five hours, expect for the large-scale dataset bigblue3 which takes 12 hours for training. The impact of using pre-trained policy is discussed in section 4.3, at lines 258-268.

---

### Official Review · Reviewer_5u46 · 2021-07-17

**Rating:** 6
**Confidence:** 4

**Summary:**

This paper proposes a joint learning method to solve the placement and routing problems together. Such problems used to be studied separately.  For the placement, this paper is also solving the placement of macro and standard cells together. Moreover, both CNN and GNN are adopted to provide different embedding model.  Finally, the experimental shows that the proposed jointly method has notable performance margin than traditional separate pipeline.

**Ethical Concerns:**

did not notice any ethical concern

**Ethics Review Area:**

["I don’t know"]

**Limitations And Societal Impact:**

did not see any negative societal impact

**Main Review:**

Pros:
	This paper is targeting an important challenge in IC design. The idea that jointly solving placement and routing is promising solution since they are strongly coupled. And the placement’s quality will severely impact the routing performance.
	The related work section is exhaustive which gives the reader a good understanding of the existing works about placement and routing. Not only the traditional methods but also the recently learning-based methods.
	In the Methods part, the author clearly described the proposed method including the problem formulation, reward design, how to calculate wirelength and congestion,  how to utilize both CNN and GNN, and etc.

Cons:
	The major issues are lack of details and insufficient experiment.
	In the experiment, only 2 circuits are used to test the proposed method. The author shows the plot or table but failed to explain the reason behind the result. For example, the author mentioned pre-trained for 300 iterations achieves the best result. But why it achieves best and why only it has a big jump around 100K timesteps?  It is better if the author can re-organize the result, present them more clearly and analyze them in detail not only just show them.

	Minor issues:
	Some abbreviations are not explained such as DRL(Line 37, Page 2), SA(Line 66, Page 3). Although reader can guess their meaning, but it may cause unnecessary mis-leading.

	The method section should be explained more than it is. Every sub-section only have one paragraph for how it is design. It is better if the author can explain why you design it like this? and how it works in the whole system?. And this will also help reader to follow your main story.


---raised my score to 6 after authors rebuttal

**Time Spent Reviewing:**

2 hours

---

> ### Author Response · Authors · 2021-08-10
> **Rebuttal to Reviewer 5u46**
>
> We are thankful for your recogniton to our work's promising research direction for joint learning of two coupled task, the exhausitive literature review, and the technical clarity. Below we respond to your comments and we do hope you would re-consider the rating given our new positive results in below.
>
> **Q1. *".. only 2 circuits are used to test the proposed method."***
>
> **Re**: We add new results of 3 additional circuits for ISPD-2005, including one large-scale dataset (bigblue3) that can make it more convincing. So far we have tested 5 out of 8 chips in ISPD-2005, and the remaining 3 chips will also be provided in the next few days if permited. The additional experimental results are shown in the tables below.
>
> First we demonstrate the statistics of three additional circuits from ISPD-2005 benchmark.
>
> | Circuits | # Total Cells | # Nets | # Movable Macros | # Fixed Macros | Design Density |
> |----------|---------------|--------|------------------|----------------|----------------|
> | adaptec3 | 451K          | 466K   | 710              | 13             | 74.5%          |
> | bigblue1 | 278K          | 284K   | 551              | 9              | 54.2%          |
> | bigblue3 | 1097K         | 1123K  | 1227             | 66             | 85.6%          |
>
> In table below we show results on Joint Macro/Standard cell placement, which is extension to Table 2. Note that DREAMPlace fails to meet congestion criteria in circuits bigblue1 and bigblue3 for overlap between macros, our approach and sequential learning placer consistently generate viable placements, which shows the advantage of our RL agent for macro placement.
>
> | Circuits | Our End-to-End Approach | Sequential Learning Placer (our immediate improvement to [1]) | Analytic Placer DREAMPlace |
> |----------|-------------------------|----------------------------|----------------------------|
> | adaptec3 | **241072304.0**             | 257956784.0                | 264789408.0                |
> | bigblue1 | **140435296.0**             | 168456768.0                | - (failed in legality)     |
> | bigblue3 | **450633360.0**             | 460581568.0                | - (failed in legality)     |
>
> [1] Mirhoseini, Azalia, Anna Goldie, Mustafa Yazgan, Joe Wenjie Jiang, Ebrahim Songhori, Shen Wang, Young-Joon Lee et al. "A graph placement methodology for fast chip design." Nature 594, no. 7862 (2021): 207-212. (Note we cited its arxiv version in our paper as at that time the nature version has not been published yet)
>
> Finally we present the results on Joint Placement and Routing, which is extension to Table 3.
>
> | Circuits  | Our Joint Learning Approach | Separate Pipeline |
> |----------|-----------------------------|-------------------|
> | adaptec3 | **32839**                       | 37029             |
> | bigblue1 | **8602**                        | 9078              |
> | bigblue3 | **94083**                       | 98480             |
>
>
> **Q2. *".. the author mentioned pre-trained for 300 iterations achieves the best result. But why it achieves best and why only it has a big jump around 100K timesteps?"***
>
> **Re**: Thanks for your important question. We have given explantion at lines 237-241 and 261-268 in the paper and will make it more clear in the new version.
>
> For the first question, we can divide it into two parts: why pre-training policy performs better than training from scratch, and why further pre-training hurts the final performance.
>
> The pre-training task we design is to only focus on the macro placement, which is more consistent with the action space of our RL agent and result is good enough to provide an intermediate macro placement. On the other hand, training from scratch starts off with random weights which lead to low-quality intermediate placement, so it needs more steps for exploration and spends more runtime on gradient based optimization placer in each episode. Therefore, finetune the pre-trained policy network takes shorter time to converge and is more likely to achieve better result. This idea is similar to curriculum learning that gradually introduces more difficult task to speed up training. However, the pre-training task solves placement problem from local perspective that tries to minimize the wirelength between macros. This will not directly lead to global minimum of full placement which needs to consider both macro and standard cells. For example, it is better to place the standard cells surrounded by macros, which can not be learned from pre-training task. Thus, excessive pre-training is prone to overfitting and hurt exploration to find a global minimum. In summary, moderate pre-training will achieve the best result in our experiment. It turns out to be 300 iterations for the test case *adaptec1*.  Similar results are observed for the other tested circuits.
>
> For the second question, the reward signal consists of two objectives to balance: minimizing total wirelength and the congestion. Note that placing cells in the same net near to each other will naturally reduce the wirelength but increase congestion of adjacent grids at the same time. Moreover, it is hard for the agent to predict how placement of macro will influence the standard cells placement and the final performance. Therefore, the reward appears to be unstable in the early stage and finally goes up around 100K timesteps. The plot will vary from different test cases, so we only make an overall analysis here. Thanks again for your valuale questions.

---

### Official Review · Reviewer_WfTv · 2021-07-17

**Rating:** 6
**Confidence:** 3

**Summary:**

The paper introduces a joint placement and routing system based on deep reinforcement learning. They use two RL agents to do placement and routing and train them using a single reward function. This design and the reward structure enables them to reach closer to the global optimum. The results show that the learned policies are effective and beats the baselines they consider.

**Limitations And Societal Impact:**

Authors mention unemployment of engineers as a negative societal impact. I do not see any other major negative societal impact from this work.

**Main Review:**

# Strengths

* Joint optimisation of placement and routing. This is enabled through their careful design of the reward function that guides the optimization policies to the true target. More specifically, their use of wire length (HPWL) enables the authors to make placement decisions which are cognizant of routing output.
* Use of both a CNN and GNN to model different aspects of the input. The netlist intrinsically has a graph structure, where as the die has a 2D structure. Their use of two different NN topologies to exploit these two different structures is commendable.
* Usage of reward distillation to enable more exploration and to somewhat mitigate the sparse reward problem: authors correctly identifies that the unavailability of intermediate rewards makes the convergence of their policies slower and hence introduces random distillation to mitigate the problem somewhat.
* Adoption of pre-training (tasks separately) and then fine-tuning

# Weaknesses and questions for authors

* Some of the terminologies of the paper is not that approachable by a non-domain expert. For instance, what is a macro and a standard cell. To improve the clarity of the paper, the authors should have a relatively small background section to ground these keywords that are used over and over again in the text.
* Evaluation
  * Authors mention about ablation studies to validate the usefulness of a multi-view policy network. However, I could not find some of those results in the paper. Specifically, I would like to know how much does having both a GNN and CNN help in the final performance of the policy?
* The baselines are not explained properly. In order for a non-domain expert to appreciate the results, the baselines should be explained better. For instance, how is sequential learning placer different from DREAMPlace. A table with differences would be helpful.
* I am not familiar with the scale of the two benchmarks. Does ISPD-2005 have only 2 benchmarks?  Are these problems of the same scale as usual place and route problems encountered in normal chip design?
* Table 2 and 3 wire length statistics are of different magnitude orders. Is it because in Table 3 you only consider macros? Does that mean standard cell placement is more important than macros since it introduces more wires? Please explain.

**Time Spent Reviewing:**

3

---

> ### Author Response · Authors · 2021-08-10
> **Rebuttal to Reviewer WfTv**
>
> Thank you for the detailed feedback and recognition to our work's novelty, potential influence, and technical soundness. Below we respond to your specific comments and look foward your feedback.
>
> Responses to main concerns and questions:
>
> **Q1. *"what is a macro and a standard cell .."***
>
> **Re**: We have given explantion at lines 20-21 in the paper and will make it more clear in the new version. In general, standard cells are basic logic cells, while macros are functional blocks such as SRAMs. When considering the experiment, each circuit contains several hundred macros of larger size and millions of standard cells.
>
> **Q2. *"... how much does having both a GNN and CNN help in the final performance of the policy."***
>
> **Re**: We conduct ablation studies based on the pretraining task (which is simpler) to verify the effectiveness of our multi-view policy network. Moreover, the cost of adding a CNN by treating the circuit board as image is negeligible, compared with the GNN. We further verify the improvement of multi-view policy network and RND module in the final performance on circuit **adaptec3** from the benchmark ISPD-2005. Results are shown in the table below.
>
> |                  | Our Joint Learning Approach | w/o GNN | w/o RND | Separate Pipeline |
> |------------------|-----------------------------|---------|---------|-------------------|
> | Total wirelength | 32839                       | 34742   | 35556   | 37029             |
>
>
> **Q3. *"... how is sequential learning placer different from DREAMPlace."***
>
> **Re**: The difference is that for sequential learning placer, macros are also arranged by RL agent, while they are placed by heuristic in DREAMPlace. A structured overview for the differences is shown as follows.
>
> |                                                                                  | targets for placement |                  | learning protocol               |                |
> |----------------------------------------------------------------------------------|-----------------------|------------------|---------------------------------|----------------|
> |                                                                                  | **macro cell**            | **standard cell**    | **reward**                          | **joint learning** |
> | our method                                                                       | RL                    | back-prop by DNN | based on cell placement result  | yes            |
> | Google's Nature 2021 [1]                                                         | RL                    | NA               | based on macro placement result | no             |
> | sequential learning placer for ablation study (our immediate improvement to [1]) | RL                    | back-prop by DNN | based on macro placement result | no             |
> | DREAMPlace                                                                       | heuristic             | back-prop by DNN | NA                              | no             |
>
>
> [1] Mirhoseini, Azalia, Anna Goldie, Mustafa Yazgan, Joe Wenjie Jiang, Ebrahim Songhori, Shen Wang, Young-Joon Lee et al. "A graph placement methodology for fast chip design." Nature 594, no. 7862 (2021): 207-212. (Note we cited its arxiv version in our paper as at that time the nature version has not been published yet)
>
> **Q4. *"Does ISPD-2005 have only 2 benchmarks? Are these problems of the same scale as usual place and route problems encountered in normal chip design?"***
>
> **Re**: The very popular benchmarks ISPD-2005 contains in total 8 circuites and it is directly derived from industrial ASIC designs. Therefore, we believe the scale is similar with problems encountered in normal chip design. We add new results of 3 additional circuits for ISPD-2005, including one large-scale dataset (bigblue3). So far we have tested 5 out of 8 chips in ISPD-2005, and the remaining 3 chips will also be provided in the next few days if permited. Additional experiment results are shown in the tables below.
>
> First we demonstrate the statistics of three additional circuits from ISPD-2005 benchmark.
>
> | Circuits | # Total Cells | # Nets | # Movable Macros | # Fixed Macros | Design Density |
> |----------|---------------|--------|------------------|----------------|----------------|
> | adaptec3 | 451K          | 466K   | 710              | 13             | 74.5%          |
> | bigblue1 | 278K          | 284K   | 551              | 9              | 54.2%          |
> | bigblue3 | 1097K         | 1123K  | 1227             | 66             | 85.6%          |
>
> Below we show results on Joint Macro/Standard cell placement, which is extension to Table 2. Note that DREAMPlace fails to meet congestion criteria in circuits bigblue1 and bigblue3 for overlap between macros, our approach and sequential learning placer consistently generate viable placements, which shows the advantage of our RL agent for macro placement.
>
> | Circuits | Our End-to-End Approach | Sequential Learning Placer (our immediate improvement to [1]) | Analytic Placer DREAMPlace |
> |----------|-------------------------|----------------------------|----------------------------|
> | adaptec3 | **241072304.0**             | 257956784.0                | 264789408.0                |
> | bigblue1 | **140435296.0**             | 168456768.0                | - (failed in legality)     |
> | bigblue3 | **450633360.0**             | 460581568.0                | - (failed in legality)     |
>
> Finally we present the results on Joint Placement and Routing, which is extension to Table 3.
>
> | Circuits  | Our Joint Learning Approach | Separate Pipeline |
> |----------|-----------------------------|-------------------|
> | adaptec3 | **32839**                       | 37029             |
> | bigblue1 | **8602**                        | 9078              |
> | bigblue3 | **94083**                       | 98480             |
>
>
> **Q5. *"Table 2 and 3 wire length statistics are of different magnitude orders. Is it because in Table 3 you only consider macros? Does that mean standard cell placement is more important than macros since it introduces more wires?"***
>
> **Re**: Yes, we only consider macros in Table 3 which evaluates the performance of joint placement and routing. For which is more important for standard cell and macro, we think macro is also very important becuase the macros are first placed followed by standard cells and their placement quality can largely influence the standard cell placement and the final performance. Exactly for this reason, in this paper, we are motivated to develop a **joint learning** approach which is new in literature.

---

### Author Response · Authors · 2021-08-21
**Additional Experimental Results**

We finish the remaining three chips in the ISPD-2005 benchmark suites and summarize their statistics as follows.

| Circuits | # Total Cells | # Nets | # Movable Macros | # Fixed Macros | Design Density |
|----------|---------------|--------|------------------|----------------|----------------|
| adaptec4 | 496K          | 516K   | 1309             | 20             | 62.7%          |
| bigblue2 | 558K          | 577K   | 948              | 22136          | 37.9%          |
| bigblue4 | 2177K         | 2230K  | 659              | 7511           | 44.3%          |

In table below we show results on Joint Macro/Standard cell placement, which is extension to Table 2. Note that DREAMPlace fails to meet congestion criteria in all circuits here due to the increasing complexity of macro placement.

| Circuits | Our End-to-End Approach | Sequential Learning Placer (our immediate improvement to [1]) | Analytic Placer DREAMPlace |
|----------|-------------------------|---------------------------------------------------------------|----------------------------|
| adaptec4 | **236391936.0**             | 255770672.0                                                   | - (failed in legality)     |
| bigblue2 | **140465488.0**             | 142037456.0                                                   | - (failed in legality)     |
| bigblue4 | **951984128.0**             | 1039976064.0                                                  | - (failed in legality)     |

Finally we present the results on Joint Placement and Routing, which is extension to Table 3.

| Circuits | Our Joint Learning Approach | Separate Pipeline |
|----------|-----------------------------|-------------------|
| adaptec4 | **63560**                      | 79573             |
| bigblue2 | **16819**                       | 17849             |
| bigblue4 | **17585**                       | 17829             |

---

### Author Response · Authors · 2021-08-31
**Waiting for Reply**

During the discussion period, the issue of insufficient experiment can be addressed by our additional experiments that cover all the chips in the ISPD-2005 benchmark suites. Moreover, further explantion for experimental details is provided in our rebuttal. We hope these additional evidences can solve most of your concerns in order to re-consider the rating.

We are looking forward to receiving comments from reviewer 5u46.

---

### Public Comment · ~Jerry_Erler1 · 2023-05-22
**A research topic or a paper**

"Joint Learning for Solving Placement and Routing in Chip Design" refers to a research topic or a paper that explores the application of machine learning techniques to address the challenges of placement and routing in chip design.

Placement and routing are critical stages in the process of designing integrated circuits (chips). Placement involves determining the optimal locations for the different components on a chip, while routing involves establishing the interconnections between these components. Both tasks are complex and time-consuming, requiring careful optimization to achieve optimal chip performance and functionality.

In recent years, researchers have been exploring the potential of machine learning algorithms to improve the efficiency and effectiveness of placement and routing. By leveraging large amounts of historical chip design data, machine learning models can learn patterns, optimize placement strategies, and generate efficient routing solutions.

The concept of joint learning in this context refers to the integration of placement and routing tasks within a unified machine-learning framework. Traditionally, these tasks were addressed separately, but by jointly considering placement and routing, it is possible to capture the interdependencies between them and generate more optimized solutions.

The research in this field focuses on developing novel machine-learning models, algorithms, and optimization techniques that can jointly solve placement and routing problems. The goal is to improve chip design efficiency, reduce design iterations, and enhance overall chip performance.

It's important to note that without specific details or context about a particular paper or research study on this topic, it is challenging to provide more specific information. If you have any particular questions or would like to explore a specific aspect of joint learning for placement and routing further, please provide more details, and I'll be happy to assist you or visit https://mipacademy.in if you are interesting to know on digital marketing. .

---

### Decision · Program_Chairs · 2021-09-27

**Decision:**

Accept (Poster)

**Comment:**

There was a consensus among reviewers that this paper should be accepted. It is the first to propose an RL agent for the combined task of placement and routing in chip design. The main critique after initial reviews were insufficient experiments. However, this was addressed to the reviewers' satisfaction by extensive additional experiments in the rebuttal. Hence, I recommentd acceptance of this paper.